# Effects of Salinity on Earthworms and the Product During Vermicomposting of Kitchen Wastes

**DOI:** 10.3390/ijerph16234737

**Published:** 2019-11-27

**Authors:** Zexuan Wu, Bangyi Yin, Xu Song, Jiangping Qiu, Linkui Cao, Qi Zhao

**Affiliations:** School of Agriculture and Biology, Shanghai Jiao Tong University, Shanghai 200240, China; wuzexuan@sjtu.edu.cn (Z.W.); bangyi2018@sjtu.edu.cn (B.Y.); sistercon@sjtu.edu.cn (X.S.); jpq@sjtu.edu.cn (J.Q.); clk@sjtu.edu.cn (L.C.)

**Keywords:** environmental, vermicomposting, kitchen wastes, salinity, health

## Abstract

Population growth and social changes have recently contributed to an exaggerated increase in kitchen wastes in China. Vermicomposting has recently been recognized as an effective and eco-friendly method of organic waste treatment through the combination of earthworms and microbes. However, the influence of salt in kitchen wastes on vermicomposting have been unknown. The goal of this study was to analyze the influence of different salinities on earthworms (*Eisenia fetida*) and the products during the vermicomposting of kitchen wastes. In our research, kitchen wastes were divided into four different salinities: 0% (A), 0.1% (B), 0.2% (C) and 0.3% (D). The chemical characters of substrates and earthworm growth were measured on the 14th day and the 28th day of composting. Our results show that the high salinity (measured >0.2%) prevented earthworms from properly growing and had negative effects on quality of products in composting. T2 (measured salinity = 0.2%) had the highest average body weight, nitrate nitrogen, and available phosphorus. Thus, the salinity of kitchen wastes should be pretreated to less than 0.2% before vermicomposting.

## 1. Introduction

Population growth and social changes have contributed to a great increase in kitchen waste, which has received public attention due to its quantity, odor and the potential for pathogenic microbial contamination [1,2]. Nearly 97 million tons of kitchen wastes were made in China in 2006 [3]. Kitchen wastes are rich in organic matter and water content. However, kitchen wastes have a low calorific value. Not only are they hard to collect and transport, but they can also lead to severe environmental problems if they are untreated before composting [4]. Meanwhile, a high salinity is associated with inhibitions and failures of fermentation processes [5]. For instance, when the salinity of a substrate is higher than 0.5%, its biodegradation efficiency will be significantly reduced in anaerobic digestion [6]. Moreover, Na^+^ reduces the activity of microorganisms and interferes with their metabolism if the salinity is high [7]. The main methods to treat kitchen waste are landfill, incineration, anaerobic digestion and aerobic composting [2,8,9], which are generally expensive, multi-step, and easily result in secondary pollution.

Vermicomposting has been recognized as a method of organic waste treatment through the combination of earthworms and microbes [10,11]. It is broadly deemed as a simple, efficient and eco-friendly technique for converting organic waste materials to high nutritional and humified casts [12]. After vermicomposting, a more fine and homogeneous product is gained compared to classical composting [13]. Shak et al. decomposed rice and cow dung with earthworms for the bio-transformation of wastes into organic fertilizer [14]. Their results showed that the available phosphorus increased by 1.2–7.3% after 60 days. The total nitrogen, potassium and phosphorus increased from 0.84% to 1.34%, from 0.84% to 1.34%, and from 1.27% to 1.83%, respectively, in a 60-day vermicomposting with wheat straw and cow dung mixture.

Vermicomposting has shown the highest total nitrogen compared to other treatments of pig dung, cow dung and rotting foliage mixtures [15].However, the salinity of kitchen waste have effects on the growth and energy budget of organisms. After different salinity treatments (0–30%), the specific growth rate of *Cynoglossus semilaevis* has been shown to at first increase and then decrease as salinity increases. Furthermore, fish have shown stress reactions such as anxiety, restlessness, and migratory behavior in unsuitable salinity conditions [16]. Elefsiniotis et al. also pointed out that the salt is toxic to bacteria, and a high salt concentration dehydrates cells due to osmotic pressure [17]. However, Hu et al. found that the excretion rate of the ammonia nitrogen of fish increased when salinity increased [18]. Boeuf et al. also pointed out that salt can improve the appetite of fish [19]. Up to now, in order to avoid the effect of salt on the kitchen waste, waste has been washed before vermicomposting in most studies [20]. Thus, the effect of salinity on earthworms was ignored. The goal of this research was to study the influence of salinity on products and earthworms during the vermicomposting of kitchen wastes.

## 2. Material and Methods

### 2.1. Organic Waste Collection and Earthworm Culture

We took the soil from the Xiaohonglou farm in School of Agriculture and Biology, Shanghai Jiao Tong University. All kitchen waste used in these experiments was produced in our lab. Salt (refined iodized salt, Zhongyan; 99.1% NaCl) was used in the experiment. In accordance with the pre-investigation of our group (data unpublished), the composition of kitchen wastes in Shanghai was: vegetable: rice: meat = 7:2:1. We kept the lipid at 5% in all treatments. [21]. The physical and chemical characters of the substrates are given in Table 1.

The earthworms (*Eisenia fetida*) were provided by Zhang Fei Company of Wuxi City, Jiangsu Province. We did not feed them for 48 h to avoid the effects of previous foods on this experiment.

### 2.2. The Vermicomposting of Feedstocks

We set four different salinities of kitchen waste in our experiment. A green-plastic pot with dimensions of 10 × 12 × 15 cm was used as the earthworm bin. Each basin was covered with a high-dense gauze and had small holes at the bottom to prevent earthworms from escaping and asphyxia from occurring (Figure 1a).

400 g of substrates (soil: kitchen wastes = 4:1) were put in each pot with 8 sexually matured earthworms. The kitchen wastes with 4 different salinities (manually added) of 0% (T1), 0.1% (T2), 0.2% (T3) and 0.3% (T4) were regarded as different groups. Each group was made 6 biological replications. The vermicomposting lasted for 28 days in an environment with 22 ± 1 °C and 65 ± 5% moisture content by injecting sterile water. The meat, vegetables and rice in the kitchen waste inherently contained salt in addition to what was exogenously added. The salinity was determined as follows: 0.1% (T1), 0.2% (T2), 0.27% (T3), and 0.39% (T4) (Table 2).

After 28 days of vermicomposting, all the earthworms except cocoons were taken out of the bin. We enumerated the earthworms one after another. After counting, the earthworms were washed in deionized water and weighed.

### 2.3. Chemical Analysis

The total organic carbon was measured by the potassium dichromate method [21]. The total nitrogen was determined by using Kay’s method [22]. The total phosphorous and available phosphorous were determined by ammonium vanadium molybdate colorimetry. The total potassium and available potassium measurements were attained via a flame photometer [23]. NH_4_^+^–N and NO_3_^−^–N were determined by an auto flow analyzer. Lipid was determined by acid titration (GB5009.229-2016). Salt was measured with silver nitrate titration (GB5009.44-2016).

### 2.4. Statistical Analysis

We reported the mean together with the standard deviation. The probability levels carried out for statistical significance were *p* < 0.05 (LSD) for the test by a one-way analysis of variance (ANOVA) using SPSS16.0. (IBM, Armonk, NY, USA). Each variable was determined for correlation (Spearman) [24] A redundancy analysis (RDA) was analyzed by the Canoco 5.0 software. (Microcomputer Power, Ithaca, NY, USA).

## 3. Results

### 3.1. Growth of Earthworms

From the 14th day to the 28th day, the average body weight (total biomass/total number) of *E. fetida* increased at first then decreased, except for T2. On the 28th day, the average body weight increased and then decreased when the salinity increased (T1–T4) (Figure 2). In the end of the vermicomposting (28th day), the average body weight (abw) of *Eisenia fetida* was higher in T2 (0.58 ± 0.04 g) and was followed by T1 (0.54 ± 0.05g). These values were significantly higher (T1) or slightly higher (T2) than T3 (0.51 ± 0.02 g) and T4 (0.50 ± 0.01 g). Though the abw of *E*. *fetida* increased by 19% in T2, the growth rate of *E*. *fetida* decreased 93% from the 14th to the 28th day. Compared to the initial abw of *E*. *fetida* in T1, T3 and T4, the abw increased by 4% at first then decreased by 2% in T3. However, it continually reduced by 4% and 22%, respectively, in T1 and T4.

### 3.2. TOC and C/N Ratio

T1–T4 decomposed the total organic carbon (TOC) by 29–56% after 28 days (Table 3). The TOC of T3 and T4 were significantly higher than that of other two treatments (T1 and T2) (*p* < 0.05). The C/N ratio of substrates presented a drastic change during vermicomposting. The final decreases of the C/N ratio of T1–T4 were in the range of 58–74% by the end of the vermicomposting. In addition, the highest decrease in the C/N ratio was in T2 which was 38% less than T4 on the 28th day.

### 3.3. TN, TP and TK

Figure 3a shows the variation of the total nitrogen (TN) in T1–T4 during the whole period of composting. Though the TN in T2 was significantly higher than that in other treatments on the 14th day, it raised around 71% among four treatments at the end of vermicomposting.

The total phosphorus (TP) increased by 2–8% in T1–T4 on the 28th day (Figure 3b). Though no significant difference appeared among four treatments, the TP in T2 and T3 was higher (*p* > 0.05) than that in T1–T4. The total potassium (TK) decreased gradually from day 0 to day 14; however, there was minimal change from day 14 to day 28. Moreover, there was no significant difference (*p* > 0.05) among the four treatments (Figure 3c).

### 3.4. NH_4_^+^–N, NO_3_^−^–N, AP and AK

Data show that the NH_4_^+^–N of the substrates in T2 was significantly higher (*p* < 0.05) than others (Figure 4a). The NH_4_^+^–N of substrates in T1 and T2 increased 22- and 34-fold on the 14th day at first, and then it reduced by 33% and 28% on the 28th day, respectively, compared to the initial substrate (10.12 mg·kg^−1^). However, the NH_4_^+^–N of T3 and T4 increased continually by 372% and 406%, respectively, at the end of the experiment.

The highest increase in NO_3_^−^–N was in T2 (38.9 g·kg^−1^), followed by T1 (36.4 g·kg^−1^), T3 (24.2 g·kg^−1^) and T4 (16.9 g·kg^−1^) (Figure 4b). With the increasing salinity, NO_3_^−^–N decreased except for T2 (*p* > 0.05). T2 presented the highest NO_3_–N, which was 2.3-fold higher than that in T4 by the end of vermicomposting (28th day). In T1–T4, NO_3_^−^–N increased by 13.5-, 14.4-, 8.9- and 6.3-fold, respectively, compared to the initial value on the 28th day.

It was found that T2 had the highest available phosphorus (AP) among four different treatments (Figure 4c) (*p* < 0.05), whether on the 14th or the 28th day. There were 59–110% increases among treatments T1–T4 compared to initial value (35.8 mg·kg^−1^).

As shown in Figure 4d, the final increases of the available potassium (AK) of T1–T4 were in the range of 2.2–2.6-fold in the end of the vermicomposting, but there was no significant difference among them. The highest and lowest values appeared in T1 (702.1 ± 32.34 mg·kg^−1^) and T2 (601.83 ± 72.87 mg·kg^−1^), respectively, by the end of vermicomposting.

### 3.5. Effects of Chemical Characteristics on the Growth of Earthworms

The relationship between the growth of *E. fetida* and the chemical characteristics of substrates was attained by a redundancy analysis (RDA) (Figure 5). Our results show that first two sorting axis Eigenvalues could explain a total of 96% of the whole variables. It could be found that the C/N ratio and NH_4_^+^–N had the most distinct influence (r = 0.96) (Table 4). Moreover, Table 4 demonstrates that NO_3_^−^–N had a high co-linearity with the average body weight of *E*. *fetida* (r = 0.7). At the same time, NO_3_^−^–N and NH_4_^+^–N were negatively correlated (r = −0.80).

RDA clearly divided our four treatments along the first sorting axis (Figure 5). Samples 1–3, 4–6, 7–9, and 10–12 represented T1, T2, T3 and T4, respectively. In Figure 5 and Table 4, we could also find that the abw of earthworms presented a significant positive correlation with NO_3_^−^–N, but they presented a significant negative correlation with the C/N ratio, the TOC and NH_4_^+^–N (*p* < 0.05).

## 4. Discussion

Data showed that the average body weight (abw) of *E*. *fetida* increased at first and then decreased as the salinity increased. It was the highest in T2 after 28 days. However, the growth rate of earthworms in T2 decreased after 14 days. The reason for the decreasing growth ability is likely that the salinity was too high for *E*. fetida to grow normally [25]. Boeuf et al. found that salt can improve the appetite of fish [19]. It is hypothesized that the salinity of kitchen waste less than 0.2% could also improve the appetite of earthworms. Furthermore, the permeation regulation of the earthworm body cannot maintain the balance of osmotic pressure between the body and the environment when in the presence of high salinity, resulting in cell water loss or expansion [26]. Earthworms would present stress responses such as restlessness and escapement under unsuitable salt conditions (salt concentration >0.2%).

The obvious reduction in the C/N ratio during vermicomposting might be due to the consumption of carbon through basic breathing and decomposition by guts [27]. Firstly, earthworms can add mucus, nitrogen-containing excretions, to substrates, which can increase the nitrogen [28]. Secondly, the TN can be fixed by free nitrogen-fixing microorganisms [29]. The C/N ratio is smaller with a higher fertility [30]. Meanwhile, salt can inhibit the degradation of organic matter [31] and the basic breathing of microbes has been shown to decrease with the increasing of salinity [32]. Moreover, when salinity is less than 0.2%, it can increase earthworm appetite to lead to more kitchen waste consumption rather than affecting the permeation pressure of earthworms. Thus, the lowest C/N ratio and the most abundant nutrients appeared in T2. In addition, the pH (6.58–7.46) of all the treatments was suitable for *E*. *fetida*. This suggests that the salt mainly affects the feeding behaviors and life activities of *E*. *fetida*.

NH_4_^+^–N and NO_3_^−^–N increased in four treatments during vermicomposting. Hu et al. also found that the excretion rate of NH_4_^+^–N of fish increased gradually with the increasing of salinity [18]. This might be due to the increase in activity of ammonifying and nitrifying bacteria during the decomposition of kitchen wastes [33]. First, earthworms decompose the organic materials, leading to the formation of NH_4_^+^–N under the action of microorganisms [34,35]. Second, NH_4_^+^–N is converted to NO_3_^−^–N (in the form of nitrate) by nitrification [36]. As for our research, when the salinity was less than 0.2%, NH_4_^+^–N increased at first then decreased. However, when the salinity was higher than 0.2%, NH_4_^+^–N gradually increased during the vermicomposting. Salt is the most important factor in determining the global patterns of microbial distribution [37]. When the salinity was lower than 0.2%, it could stimulate the biological excretion of NH_4_^+^–N. Nevertheless, when salinity is higher, the microbe’s cells lose water to reduce the conversion of ammonia nitrogen and nitrate nitrogen pressure [17].

After the decomposition of kitchen wastes, available phosphorus has been found to increase in all treatments [38,39]. The increase in the AP might be due to the presence of large amount of microbes in the gut of earthworms and enzymatic reactions in the process of vermicomposting [40]. Adi and Noor [41] found that the P-mineralization might be due to the acid phosphatase enzyme that is present in one and a half months of vermicomposting [42]. The activity of gut phosphatases in *E*. *fetida* has been found to contribute to the mineralization and mobilization of phosphorus [43]. The fastest growth of earthworms in T2 (measured salinity: 0.2%) might have been due to the highest intestinal phosphatase activities of the earthworms. Salt plays an important role in enzyme reaction, maintaining biofilm balance and regulating permeation pressure during the growth of microorganisms. However, high salt concentrations could also inhibit enzyme catalysis. The activity of most soil redox enzymes significantly decreased with the increasing of salinity [43,44]. Therefore, when salinity was more than 0.2%, as the salinity increased, the AP was significantly reduced (*p* < 0.05).

## 5. Conclusions

The kitchen waste with 0.2% measured salinity (T2) had the highest average body weight of earthworms and nutrients (NO_3_^−^–N and AP) (*p* < 0.05). This may have happened because: (1) Proper salinity (<0.2%) can improve the appetite of earthworms; (2) the permeation regulation of the earthworm body cannot maintain the balance of osmotic pressure between the body and the environment in high salinity condition, resulting in cell water loss or expansion so as to not survive normally; (3) a high salinity (>0.2%) can lead to cell dehydration due to an imbalance in osmotic pressure. In conclusion, the salinity of kitchen wastes should be pretreated to less than 0.2% before vermicomposting.

## Figures and Tables

**Figure 1 ijerph-16-04737-f001:**
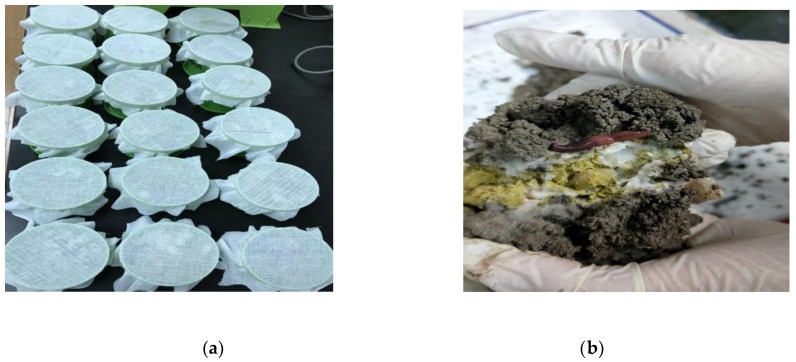
Vermicomposting device (**a**) and the earthworm (*Eisenia fetida*) within the kitchen wastes (**b**).

**Figure 2 ijerph-16-04737-f002:**
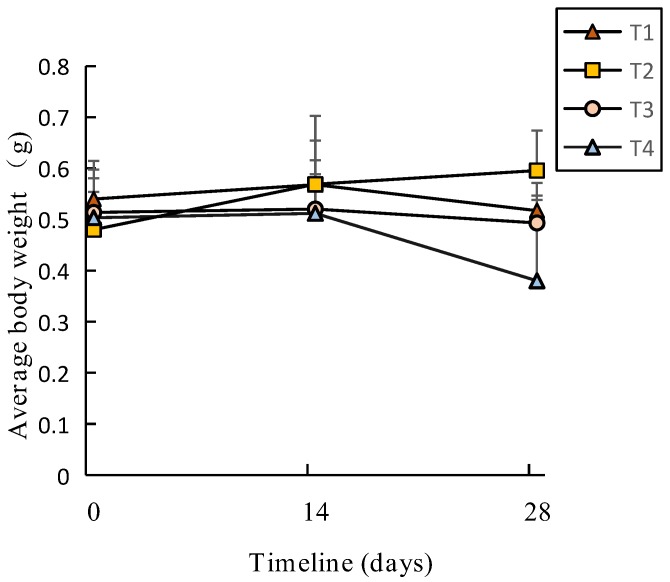
The average body weight of *E. fetida* during the vermicomposting.

**Figure 3 ijerph-16-04737-f003:**
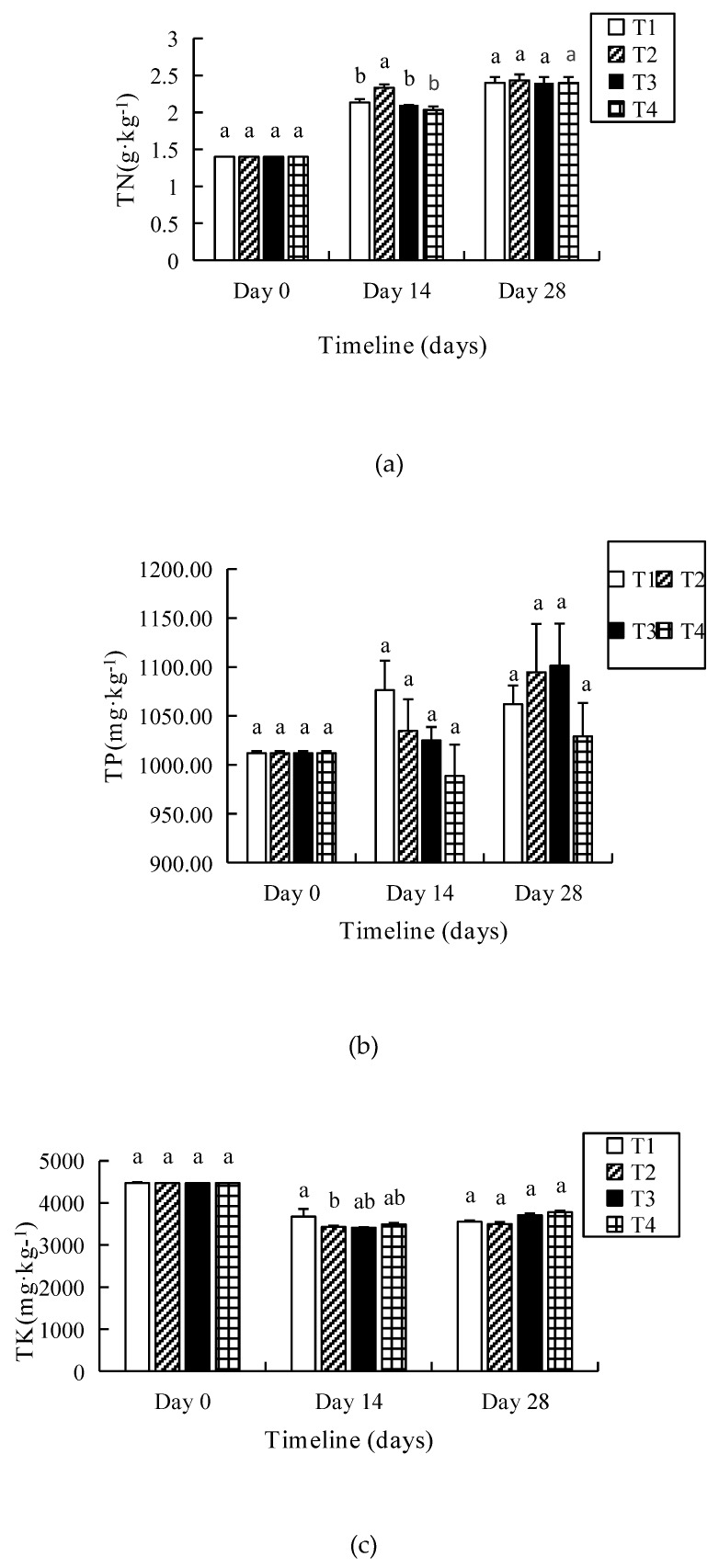
The total nitrogen (TN) (**a**), total phosphorus (TP) (**b**), and total potassium (TK) (**c**) of substrates in different salinity treatments during vermicomposting. The significant difference (*p* < 0.05) is indicated by different letters.

**Figure 4 ijerph-16-04737-f004:**
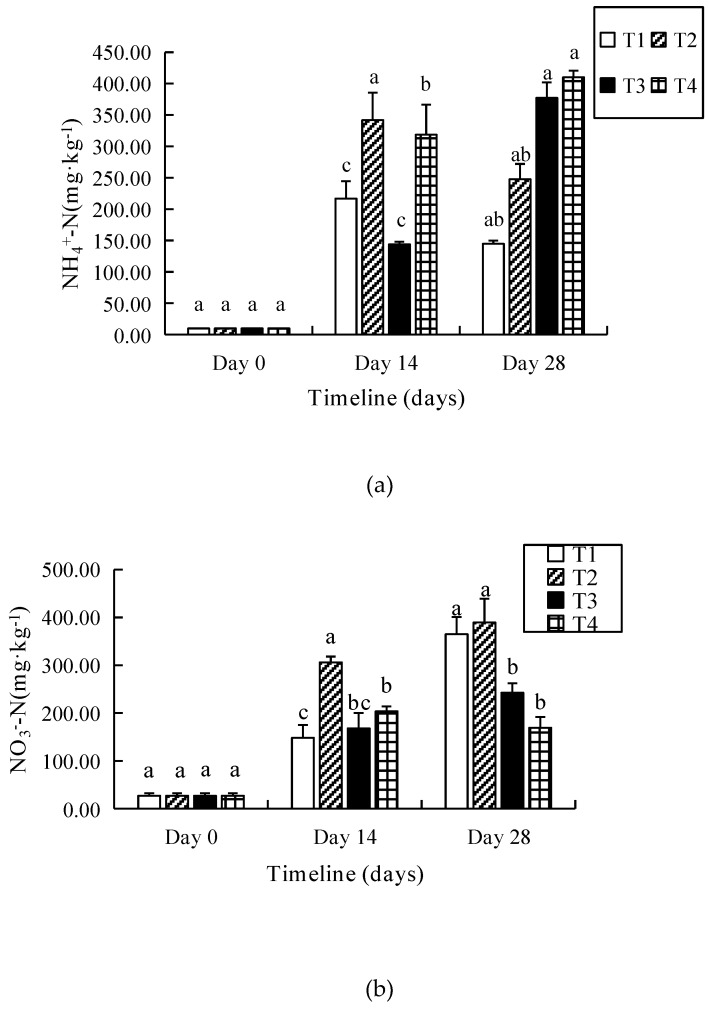
NH_4_^+^–N (**a**), NO_3_^−^–N (**b**), available phosphorus (AP) (**c**) and available potassium (AK) (**d**) of substrates in different salinity treatments during vermicomposting.

**Figure 5 ijerph-16-04737-f005:**
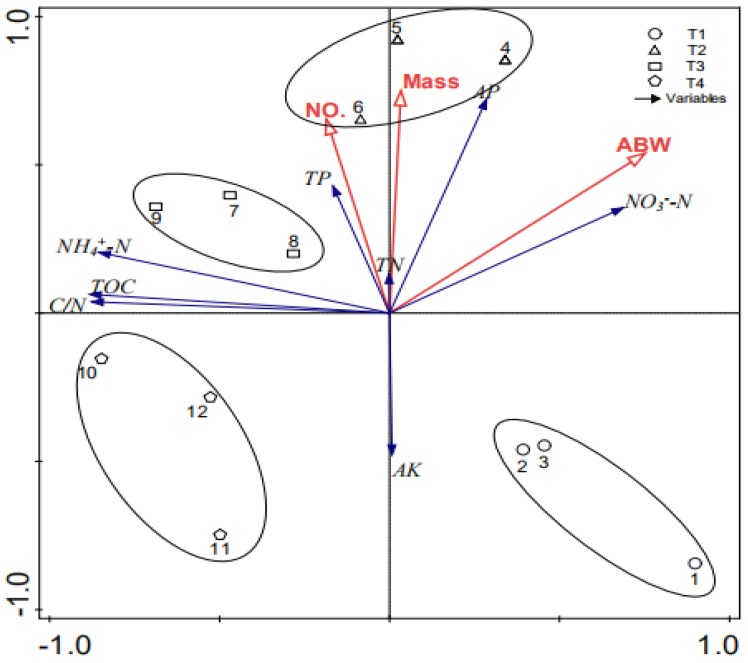
Redundancy analysis (RDA) between the chemical characteristics of substrates and the culture of *E*. *fetida*.

**Table 1 ijerph-16-04737-t001:** Physical and chemical properties of soil and kitchen wastes.

	Index (g·kg^−1^)	TOC	TN	NH_4_^+^-N	NO_3_^−^–N	TP	TK	AP	AK	pH
**substrates**										
Soil		37.4	1.40	0.01	0.03	1.01	4.40	0.067	0.270	7.60
Kitchen wastes		506	60.0	0.01	0.17	5.50	16.2	3.30	15.5	8.50

(TOC: total organic carbon, TN: total nitrogen, NH_4_^+^–N: ammonium nitrogen, NO_3_^−^–N: nitrate nitrogen, TK: total potassium, AP: available phosphorus, TP: total phosphorus, AK: available potassium).

**Table 2 ijerph-16-04737-t002:** Salinity of kitchen wastes.

Treatments	Setting Value (%)	Measured Value (%)
T1	0	0.1
T2	0.1	0.2
T3	0.2	0.27
T4	0.3	0.39

**Table 3 ijerph-16-04737-t003:** The total organic carbon (TOC) and C/N ratio of substrates during the vermicomposting (mean ± SD, n = 3).

Treatments	Total Organic Carbon (g·kg^−1^)			Biodegradation Rate %	C/N		
	Day 0 (Soil)	Day 14	Day 28		Day 0 (Soil)	Day 14	Day 28
T1	37.42 ± 0.44	33.27 ± 0.15 ^a^	19.70 ± 1.30 ^b^	47%	26.73 ± 0.6	14.27 ± 0.35 ^c^	8.10 ± 0.56 ^b^
T2	37.42 ± 0.44	28.23 ± 1.46 ^b^	16.52 ± 1.62 ^b^	56%	26.73 ± 0.6	13.25 ± 0.91 ^c^	6.89 ± 0.66 ^b^
T3	37.42 ± 0.44	34.53 ± 0.97 ^a^	24.36 ± 1.73 ^a^	35%	26.73 ± 0.6	16.44 ± 0.46 ^b^	10.14 ± 0.49 ^a^
T4	37.42 ± 0.44	36.42 ± 0.15 ^a^	26.48 ± 0.75 ^a^	29%	26.73 ± 0.6	17.92 ± 0.38 ^a^	11.05 ± 0.66 ^a^

Different letters indicate significant differences among treatments (one-way ANOVA, followed by the Turkey’s *t*-test (*p* < 0.05)).

**Table 4 ijerph-16-04737-t004:** Correlation coefficient of chemical characteristics of substrates and culture of *E*. *fetida.*

	TOC	TN	TP	AP	AK	NH_4_^+^–N	NO_3_^−^–N	C/N	No.	Biomass	Abw
TOC	1										
TN	0.02	1									
TP	−0.04	0.09	1								
AP	−0.4	0.19	0.54	1							
AK	−0.3	0.22	0.31	−0.07	1						
NH_4_^+^–N	0.97 **	−0.01	0.01	−0.3	−0.33	1					
NO_3_^−^–N	−0.81 **	0.15	0.47	0.74 **	0.33	−0.80 **	1				
C/N	0.99 **	−0.14	−0.05	−0.4	−0.31	0.96 **	−0.82 **	1			
^a^ No.	0.2	0.2	0.66 *	0.28	−0.18	0.23	0.13	0.15	1		
Biomass	0.39	0.20	0.63 *	0.43	0.28	0.04	0.30	−0.04	0.97 **	1	
^b^ Abw	−0.66 *	0.12	0.30	0.53	−0.13	−0.61 *	0.70 *	−0.67 *	0.50	0.69 *	1

Notes: * : significant level = 0.05 (two-tailed), ** : significant level = 0.01 (two-tailed). a: number of earthworms. b: the average body weight.

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
