# Peer review of "Effects of Salinity on Earthworms and the Product During Vermicomposting of Kitchen Wastes"

_ijerph, 2019, doi:10.3390/ijerph16234737_

Round 1
Reviewer 1 Report
1. How did you choose this ration for the composition of kitchen wastes in Shanghai (7:2:1), please include some references or some data.
2. why did you choose only this earthworm Eisenia fetida
3. Do you think the 48hrs fasting is sufficient to clean the earthworm with previous food? include some reference for it.
4. TOC of T3 and T4 were significantly higher. Do you think is the statement right? What is a significant percentage?
5. In the introduction, part mention some information about the metabolism of an earthworm towards salinity.
Author Response
Point 1: How did you choose this ration for the composition of kitchen wastes in Shanghai (7:2:1), please include some references or some data.

Response 1: We conducted a survey of 10 districts in Shanghai in March-April 2018. A total of 300 questionnaires were collected. 60% of people have chosen the answer of vegetable: rice: meat=7:2:1. The pie chart as attachment.
Point 2: 2. why did you choose only this earthworm Eisenia fetida
Response 2: On one hand, it is fed on organic matter, and the kitchen waste is rich in organic matter. On the other hand, the Eisenia fetida is the indicator organism commonly used in international standards to treat soil pollutants and wastes (ISO).
ISO (International Standard Organization), 1998. Soil quality — Effects of pollutants on earthworms — Part 1: Determination of acute toxicity to Eisenia fetida. ISO, Geneva. Standard Number No. 11268-1:2012(en)
Point 3:Do you think the 48hrs fasting is sufficient to clean the earthworm with previous food? include some reference for it.
Response 3: Other people's experiments about vermicomposting of kitchen wastes did not consider about the effects of intestinal contents. For other 24 hours of antibiotic contamination experiments, we carried out 48 hours for the sake of insurance.
Lin D , Zhou Q , Xu Y , et al. Physiological and molecular responses of the earthworm (Eisenia fetida) to soil chlortetracycline contamination[J]. Environmental Pollution, 2012, 171(none):46---51.
Point 4: TOC of T3 and T4 were significantly higher. Do you think is the statement right? What is a significant percentage?
Response 4: we conclude that T3 and T4 are significantly higher than T1 and T2 by one-way ANOVA (p<0.05).
Point 5: In the introduction, part mention some information about the metabolism of an earthworm towards salinity.
Response 5: No one has yet studied the effect of salt on earthworms, so no relevant references have been found.
Reviewer 2 Report
The topic is important. The information is good and relevant. The English and grammar need improvement as it is difficult to read and in some places, difficult to understand. Please avoid use of "etc" - instead, state the information. The reader may not know what else there is so you need to spell it out. Some of the graphs are also a little confusing, but improvement of the language (and word usage) may help solve this issue.
Overall, I think the study should be published once the language, use of words and grammar are corrected. It is important to know how our actions (kitchen waste) affect the downstream environment (earthworms) and equally important, how we can fix it (reduce salinity).
Author Response
Point 1: The topic is important. The information is good and relevant. The English and grammar need improvement as it is difficult to read and in some places, difficult to understand. Please avoid use of "etc" - instead, state the information. The reader may not know what else there is so you need to spell it out.
Response 1: Thank you for your suggestion. Etc has been modified in the text.
Point 2: Some of the graphs are also a little confusing, but improvement of the language (and word usage) may help solve this issue. Overall, I think the study should be published once the language, use of words and grammar are corrected.
Response 2: The language, use of words and grammar have been corrected.
Point 3:It is important to know how our actions (kitchen waste) affect the downstream environment (earthworms) and equally important, how we can fix it (reduce salinity).
Response 3: Thank you for your comments. We will discuss the principle of earthworms metabolite salt in subsequent articles.
Round 2
Reviewer 2 Report
Here are the review comments that go with the highlighted document.Editorial comments on “Effects of salinity on earthworms”
Each comment corresponds to a yellow highlighted area in the document. I hope these more detailed comments are helpful.
line 15 – Salinity should be “salinities”
line 17 – presented should be ‘showed’ or ‘demonstrated”
line 27 – use either ‘however’ or ‘but’ – not both. The kitchen waste has
line 31 – “descended” is misused – “reduced” would be better
line 35 – “ … and results in secondary pollution”
Line 45 – “compared to”
Line 48 to 49 – “at first increased, then decreased as salinity increased.”
Line 53 – “increased when salinity increased”
Line 61 – “All kitchen waste used in these experiments was produced in our lab.”
Line 65 – change “were” to “are”
Line 77 – “salinities”
Line 79 – “in an environment”
Line 80 – “The meat, vegetables and rice in the kitchen waste inherently contained salt in addition to what was added exogenously.”
Line 84 – “We enumerated the earthworms”
Line 96-97 – take out “ a kind of machine called”
Line 107 – “increased at first, then decreased”
Line 113 – “increased by 4% at first, then”
Line 121 – “ by the end”
Line 126 – “showed variation”
Line 128 – “by the end” or “at the end”
Line 130 – “were” should be “was”
Line 131 – “there was minimal change” or if it was not statistically significant “there was no significant change”
Line 132 – take out “appeared”
Line 143 – change ‘firstly”
Line 149 – “by the end”
Line 157 – take out “had the” and change “in the end” to “by the end”
Line 170 – “characters” should be “characteristics”
Line 173 – “demonstrated” change to “demonstrates that”
Line 174 – “co-linearity”
Line 175 – take out “extraordinary”
Line 189 – change ‘firstly’
Line 191 – “descending’ change to “decreasing”
Line 192 – this sentence is very confusing, try “The reason for the decreasing growth ability is likely that the salinity was too high for E. fetida to growth normally.”
Line 198 – I’m not sure “escapement” is the right term?? Do you mean they try to escape?
Line 200-202 – don’t use “For one thing” or “For another thing” – this is the appropriate use for “Firstly” and “Secondly”
Line 204 – I’m not sure I understand “soil basic breathing” please try to describe better. Do you mean that the basic breathing of the organism decreased??
Line 205 – “appetizing “ change to “increase the earthworms appetite”
Line 208 – change “it might imply” to “ this suggests”
Line 212 – eliminate one “bacteria”
Line 216 – change firstly
Line 220 – “ when salinity is higher, the microbe’s cells loose water”
Lien 224 – “phosphatase of substrates” – may just want to say “enzymatic reactions”
Line 225 – “enzyme” spelling
Line 236 – “This may be caused by:”
Line 240 – “cell dehydration”

Author Response
For example:
Point 1: Line 113 – “increased by 4% at first, then”
Response 1: Thank you for your suggestion. “Compared to the initial abw of E. fetida in T1, T3 and T4, the abw increased by 4% firstly and then decreased by 2% in T3” was replaced by “Compared to the initial abw of E. fetida in T1, T3 and T4, the abw increased by 4% at first then decreased by 2% in T3.
Point 2: Line 204 – I’m not sure I understand “soil basic breathing” please try to describe better. Do you mean that the basic breathing of the organism decreased??
Response 2: Yes, I mean that the basic breathing of the organism decreased with salinity increasing. The sentence was replaced by “Meanwhile, salt can inhibit the degradation of organic matter and the basic breathing of microbes decreased with the increasing of salinity”.
Point 3:Line 198 – I’m not sure “escapement” is the right term?? Do you mean they try to escape?
Response 3: Yes. In our experiment, we found that earthworms were hiding at the bottom of the pot with the salinity increasing.
